# Vascular Endothelial Growth Factor-D (VEGF-D): An Angiogenesis Bypass in Malignant Tumors

**DOI:** 10.3390/ijms241713317

**Published:** 2023-08-28

**Authors:** Syeda Mahak Zahra Bokhari, Peter Hamar

**Affiliations:** Institute of Translational Medicine, Semmelweis University, 1094 Budapest, Hungary; syeda.bokhari@phd.semmelweis.hu

**Keywords:** VEGF-D, FIGF, angiogenesis, lymphangiogenesis, growth factors

## Abstract

Vascular endothelial growth factors (VEGFs) are the key regulators of vasculogenesis in normal and oncological development. VEGF-A is the most studied angiogenic factor secreted by malignant tumor cells under hypoxic and inflammatory stress, which made VEGF-A a rational target for anticancer therapy. However, inhibition of VEGF-A by monoclonal antibody drugs led to the upregulation of VEGF-D. VEGF-D was primarily described as a lymphangiogenic factor; however, VEGF-D’s blood angiogenic potential comparable to VEGF-A has already been demonstrated in glioblastoma and colorectal carcinoma. These findings suggested a role for VEGF-D in facilitating malignant tumor growth by bypassing the anti-VEGF-A antiangiogenic therapy. Owing to its high mitogenic ability, higher affinity for VEGFR-2, and higher expression in cancer, VEGF-D might even be a stronger angiogenic driver and, hence, a better therapeutic target than VEGF-A. In this review, we summarized the angiogenic role of VEGF-D in blood vasculogenesis and its targetability as an antiangiogenic therapy in cancer.

## 1. Introduction

In order to meet the increased nutrient and oxygen demand, growing malignant tumors require blood supply, and hence, promote angiogenesis from existing vessels by inducing sprouting [1]. The malignant tumor vasculature has a significantly different structure and physiology as compared to normal blood vessels [2]. Vascular endothelial growth factors (VEGFs) are the primary growth factors regulating angiogenesis. Upon binding their corresponding vascular endothelial growth factor receptors (VEGFRs), VEGFs promote the proliferation and migration of endothelial cells, tube formation, increase vascular permeability and vascular endothelial cell survival, altogether angiogenesis [3]. Under pathologic conditions such as malignant tumor development, poor vasculature, and hypoxia trigger the expression of VEGFs, which activate ECs both in an autocrine and paracrine fashion, thereby increasing EC proliferation and migration [4]. Secreted VEGFs also increase vascular permeability and facilitate the transfer of plasma proteins into the extracellular matrix, where these proteins provide provisional support to the incoming ECs and facilitate vasculogenesis [3,5].

VEGFs belong to the Platelet Derived Growth Factor (PDGF) family, an important mitogenic family that modulates various biochemical pathways involved in cell growth, proliferation, and survival, as well as maintaining the structural integrity of the cell [6]. VEGFs are particularly important for blood vessel formation during embryogenesis. In mammals, members of the VEGF family include VEGF-A, VEGF-B, VEGF-C, VEGF-D, and Placental Growth Factor (PIGF). There are three types of VEGF receptors to which the members of the VEGF family can bind with varying affinity and specificity, thereby eliciting different responses. VEGFR-1 and VEGFR-2 are expressed on vascular endothelial cells (VECs) and, in some instances, on non-endothelial cells [7]. VEGFR-3 is expressed particularly on lymphatic endothelial cells (LECs) [3,8,9] (Figure 1).

### 1.1. VEGF-A and Its Receptors

VEGF-A, also often referred to as VEGF, and its isoforms bind VEGFR-1 and VEGFR-2 and regulate the differentiation and development of the vascular system [7]. VEGF-A is an essential protein for embryonic development and angiogenesis in adults. VEGF-A can exist both in soluble and matrix-bound isoforms. The soluble isoform stimulates vessel enlargement, whereas the matrix-bound isoform stimulates vascular branching [13]. VEGF-A activation of VEGFR-2 leads to EC mitogenesis and enhanced vascular permeability [14]. The indispensability of VEGF-A in vasculogenesis was proven as genetic deletion of VEGF-A resulted in the disruption of vascular development and, consequently, early embryonic lethality in mice [15]. VEGF-A produced by normal ECs leads to vascular homeostasis in an autocrine fashion, while VEGF-A production by cancer cells leads to vascular sprouting and abnormal tumor angiogenesis in an autocrine fashion [16]. 

Although VEGF-A can bind both VEGFR-1 and VEGFR-2, Canonical angiogenesis is achieved by the activation of VEGFR-2, not VEGFR-1. The mechanism of VEGFR-1 signaling is not well understood, as during embryonic development, VEGFR-1 acts as a decoy and keeps VEGF-A from binding VEGFR-2. This decoy action of VEGFR-1 fine-tunes the actions of VEGF-A on stalk cells of growing blood vessels and by recruiting pericytes for vascular stability [17]. In adult life, VEGFR-1 activation by VEGF-A, PIGF, or VEGF-B stimulates inflammation by attuning activation and migration of macrophage cells [18,19]. Likewise, in ischemic and tumor tissues, VEGFR-1 can be activated by VEGF-A and promote angiogenesis, extracellular matrix invasion, and immune evasion [20,21]. The physiologic role of VEGFR-2 is the modulation of signaling pathways leading to EC proliferation, survival, migration, vascular permeability, and angiogenesis [22]. In tumor cells, VEGFR-2 activation promotes VECs’ proliferation and recruitment. Under hypoxic conditions, tumor cells produce several growth factors that influence angiogenesis, including VEGFs (mainly VEGF-A and VEGF-D) that activate VEGFR-2 and mediate angiogenesis to counteract oxygen shortage in the tumor core [20,23,24].

Furthermore, the interaction of VEGF-A with VEGFR-2 is facilitated by NeuroPilin (NP) [25]. NP 1,2 are small membrane glycoproteins originally identified in neurons with pleiotropic functions [26]. They regulate the modulation of growth factor signaling in various processes, such as development, angiogenesis (binding VEGFs), lymphangiogenesis, Immune regulation, cardiovascular disease, and cancer progression, including tumor angiogenesis, invasion, metastasis, and therapy resistance [27,28,29,30,31,32]. They are also involved in enhancing other VEGF family members’ interaction and activation of corresponding receptors [11,25,33,34]. 

### 1.2. VEGF-C and Its Receptors

All members of the VEGF family share a VEGF homology domain (VHD). The VHD is the central domain of all VEGF family members that binds the receptor. However, there are amino acid-based structural differences and additional domains that affect the VHD’s binding to the receptor. Two members of the family VEGF-C and VEGF-D have additional N- and C- terminal domains flanking the VHD and are secreted as preproproteins, later proteolytically cleaved to their mature form [8,11,12]. The flanking domains have complex modulatory roles. For example, like mature VEGF-C, prepro VEGF-C can bind VEGFR-3, but it cannot activate the receptor, thereby acting as a competitive blocker. On the other hand, VEGF-C, with its C terminal propeptide processed, can activate the Neuropilin (NP-1) co-receptor, promoting lymphangiogenesis [11,35,36]. The proteolytic cleavage of VEGF-C to its mature form can be initiated by ADAMTS3, plasmin, Cathepsin D, thrombin, and Prostate Specific Antigen (PSA) [11,37,38,39]. Different proteases can result in different mature forms of VEGF-C or VEGF-D, leading to different receptor affinity and activation potentials [11].

Although both VEGF-C and VEGF-D can bind and activate VEGFR-3, VEGF-C is the main member associated with lymphangiogenesis. The role of VEGF-D is much more subtle in lymphangiogenesis as compared to VEGF-C, as VEGF-C is crucial for embryonic lymphangiogenic growth, but VEGF-D is rather redundant. VEGF-D knockout mice experience no significant phenotypical alteration and undergo normal lymphangiogenesis [40]. Thus, the role of VEGF-C is better characterized as a lymphangiogenic factor. Nevertheless, VEGF-D can initiate lymphangiogenic sprouting in the absence of VEGF-C [41]. It, therefore, has a supporting or compensatory role in lymphangiogenesis by activating VEGFR-3.

Upon binding and activating VEGFR-3, VEGF-C promotes proliferation, migration, and survival of LECs and initiates lymphatic vessel sprouting from large veins [42,43]. The indispensability of VEGF-C was confirmed by knockout studies when VEGF-C null mice encountered pre-natal death owing to a lack of lymphatic development [42]. VEGF-C has also demonstrated a pro-lymphangiogenic role in cancer. Human breast carcinoma cells with high VEGF-C expression promoted tumor lymphangiogenesis and, hence, metastasis in a mouse model; however, this spread could be halted by trapping VEGF-C with soluble VEGFR-3 [44]. High VEGF-C expression can facilitate metastasis by increasing the overall number of lymphatic vessels and the interaction between tumor cells and LECs, thereby promoting the entry of tumor cells into the lymph vessels [45,46]. 

### 1.3. VEGF-D and Its Receptors

Vascular endothelial growth factor-D (VEGF-D) is a secreted glycoprotein that, according to its first descriptions [47], binds and activates VEGFR-3, promoting the proliferation of lymphatic endothelial cells and hence promoting lymphangiogenesis. It was identified first as a mitogenic growth factor, secreted as an effector protein of the proto-oncogene c-fos, which is an early response transcription factor involved in cell proliferation and differentiation in cultured fibroblasts. Hence, VEGF-D was originally named c-Fos-Induced Growth Factor (FIGF) [48]. In murine embryos, VEGF-D was expressed in all major organs, including the kidney, lungs, adrenal glands, vertebral column, etc. [49]. Soon after its discovery in mice, VEGF-D was characterized in humans. In healthy humans, VEGF-D expression has been demonstrated in the lungs, heart, skeletal muscle, colon, and small intestine [50].

Human VEGF-D has 85% sequence homology to its mouse counterpart. There are structural similarities with VEGF-C (48%) and, to a lesser extent, with VEGF-A (31%). As already mentioned, VEGF-D is secreted as a preproprotein, which is cleaved to its active form. Different proteases can produce different cleaved products with unique receptor-binding abilities and functions. Similarly, there is a partially processed VEGF-D form with the C-terminal flanking peptide removed, but the N-terminal flanking peptide is still attached via disulfide bridges [51]. Classically, VEGF-D is associated with lymphangiogenesis by activation of VEGFR-3 [52]. However, VEGF-D has the ability to activate various receptors and mediate complex crosstalk between them (Figure 2). Mature VEGF-D is a non-covalent dimer, which has a 290- and 40-fold higher affinity to VEGFR-2 and VEGFR-3, respectively, as compared to its long unprocessed form [47,53]. The inability to achieve the proteolytic cleavage renders VEGF-D ineffective as an angiogenic protein as the unprocessed form cannot bind VEGFR-2 [12,38]. Mature VEGF-D induces both angiogenesis and lymphangiogenesis in vitro and in vivo via binding and activating VEGFR-2 or VEGFR-3, depending on the availability of ECs expressing one or the other receptor [54]. 

Interestingly, VEGF-D is the only member of the family that has different receptor affinity in mouse and human species. VEGF-D binds and activates VEGFR-3 in mice while activating both VEGFR-2 and VEGFR-3 in humans. However, in mice, VEGF-D induces blood vessel proliferation by activating VEGFR-3 instead of VEGFR-2 [55]. Thus, despite the differences in receptor affinity, the physiologic effects (stimulation of both angio- and lymphangiogenesis) are similar. However, owing to the presence of VEGFR-3 in the blood vascular endothelial cells, VEGF-D can promote angiogenesis via VEGFR-3 activation as well [56]. Like VEGF-A, VEGF-D induces the proliferation of vascular endothelial cells, promoting the growth of VEGFR-2 positive blood vessels. The response is strong but delayed [57], which can explain why VEGF-D has not been classically affiliated with blood vessel angiogenesis. There might be a feedback loop where different receptors are upregulated in response to ligand binding; hence, VEGF members have a cooperative role in vasculogenesis.

VEGF-D also has the ability to induce VEGFR-2/3 hetero-dimerization in both vascular and lymphatic endothelial cells [58]. ProVEGF-D, with either its N or C terminal removed, has the ability to initiate VEGFR-2/VEGFR-3 heterodimerization [59]. As the heterodimer has an altered trans-phosphorylation tyrosine residue in the VEGFR-3 unit, the binding of partially or fully processed VEGF-D elicits distinct signaling and responses [59]. VEGFR-2/VEGFR-3 heterodimer activation can promote sprouting angiogenesis [58]. This ability of partially processed and fully mature VEGF-D to activate VEGFR2/VEGFR-3 heterodimer could be one of the signaling mechanisms by which VEGF-D can induce angiogenesis. 

Both mature and C-terminal proVEGF-D have the ability to bind NeuroPilin (NP) NP-1 and NP-2 [59]. Upon activation by VEGF-D, VEGFR-3 is co-localized and internalized with NP-2, which indicates that NP-2 acts as a facilitating co-receptor in VEGF-D mediated signaling [25,33,34]. VEGF-D can modulate angiogenesis in an NP-1-dependent fashion as well, with NP-1 either acting as a co-receptor for efficient receptor activation or by NP-1-mediated phosphorylation of downstream VEGFR-2 effectors in breast and other cancers [60,61]. We can, therefore, infer that the angiogenic signaling mediated by VEGF-D could be NP-dependent.
Figure 2VEGF-D mediated angiogenic signaling in lymphatic or blood capillary endothelial cells. PreproVEGF-D can bind VEGFR-3 and activate lymphangiogenesis. However, proteolytic cleavage of the preproVEGF-D provides mature VEGF-D, which can bind and activate VEGFR-3 but also VEGFR-2 and induce VEGFR-2/3 heterodimerization [47,58,59]. Created with BioRender.com accessed on 27 July 2023.
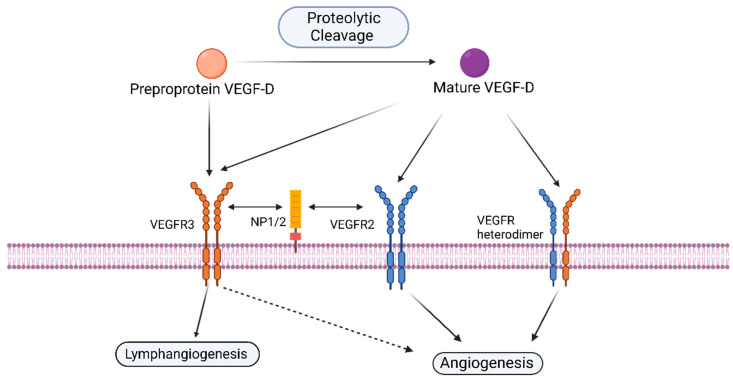



### 1.4. Other—Less Well Characterized—Members of the VEGF-Family

VEGF-B was first described in the 1990s. It is expressed mainly in muscle tissue of the heart and skeletal muscle. VEGF-B has two isoforms which can bind and activate VEGFR-1. It mainly maintains new blood vessels and prevents vascular leaking in the heart. Other physiological sites of expression include renal tubular epithelial cells [62]. VEGF-B, unlike other members of its family, is the least angiogenic factor [10,63]. However, its angiogenic potential plays a role in cardiac ischemia, where VEGF-B stimulates angiogenesis by displacing VEGF-A from VEGFR-1 [64,65]. Furthermore, a stronger expression of VEGF-B was observed in human and rodent tumor cells [64]. 

VEGF-E expressed in the Orf Parapox virus has a 25% amino acid homology to mammalian VEGF-A. The virus typically infects goats and sheep; however, it can occasionally infect humans and cause lesions [66]. In humans, VEGF-E has mitogenic effects on ECs and can enhance vascular permeability by activating VEGFR-2, similar to VEGF-A [67]. Its role in cancer is not characterized; however, its high angiogenic ability can be of therapeutic value in cardiovascular diseases [68]. 

In 1998, Komori isolated VEGF-F from the venom of *Vipera aspis aspis* (Aspic Viper), hence also termed snake venom VEGF [69]. Although not found in the human body, its potent ability to induce vascular permeability and proliferation of endothelial cells is being investigated for therapeutic interventions [69,70].

Placental Growth Factor (PIGF) expression in normal tissues is low or undetectable, and it does not affect angiogenesis directly. PIGF is important in facilitating VEGF-A binding to VEGFR-2 by binding VEGFR-1 and preventing the decoy action of VEGFR-1 [71]. However, PIGF expression is upregulated in cancer, correlating to progressive tumor grade and stage. PIGF promotes angiogenesis by activating the VEGFR1/VEGFR-2 heterodimer by dimerizing with VEGF-A itself [72]. 

## 2. Lymphangiogenesis and Angiogenesis, the Paths Cross?

Lymphangiogenesis is essentially a form of angiogenesis in which the vessels usually originate from large veins [73]. Much like blood vascular angiogenesis, lymphangiogenesis requires the activation of endothelial cells for proliferation. However, the triggers for this EC activation differ. As stated previously, blood vessel angiogenesis is usually triggered by a lack of oxygen supply, while lymph vessel growth is initiated in response to the interstitial pressure in normal development and inflammation in pathological cases [74]. Lymphatic Endothelial Cells (LECs) have an oak leaf-like morphology and are connected to each other with overlaps; however, LECs of the collecting lymphatic ducts have smooth and elongated morphology with tight junctions between cells [75]. 

Physiologically, lymphatic vessels differ from normal blood vessels as they lack tight junctions, and the LECs are connected by overlaps that can mechanically open to cope with increased interstitial pressure. Furthermore, lymph vessels lack a complete basement membrane [73,76]. Notably, this physiological profile is like the physiology of tumor vasculature [77,78,79,80,81]. 

In cancer, angiogenesis and lymphangiogenesis rely on a complex interplay of secreted growth factors and cytokines, a potential crosstalk between the two processes [82]. Canonically, LECs have a high expression of VEGFR-3, and their activation by VEGF-C or VEGF-D initiates and maintains lymphangiogenesis [83]. However, VEGF-A can also contribute to lymphangiogenesis by activating VEGFR2 and/or VEGFR2/VEGFR3 heterodimer and indirectly by recruiting VEGF-C and VEGF-D secreting macrophages and mast cells [84]. However, the contribution of VEGF-A is negligible compared to VEGF-C/D in inducing lymphangiogenesis [83]. 

It is possible that the pathways cross or mimic each other in cancer angiogenesis, although yet unidentified, exploring this relationship might open new directions to cancer metastasis and drug delivery research. 

## 3. Angiogenesis in Cancer and Its Therapeutic Targeting

A tumor cell’s ability to induce angiogenesis is one of the hallmarks of cancer, as angiogenesis is essential for tumor cells to survive [85]. Furthermore, the autocrine production of growth factors and cytokines by the active ECs promotes tumor progression [86]. Tumor angiogenesis (the angiogenic switch) is initiated in response to hypoxic or inflammatory stimuli in the tumor microenvironment. During the angiogenic switch, ECs are activated and start to produce angiogenic growth factors such as Fibroblast Growth Factors (FGFs), TGF-β, PDGF, and VEGFs (A and D) [24,87]. The binding of these growth factors to their EC receptors leads to the production of matrix-degrading enzymes, enabling blood vessel growth. VEGF-A mediated activation of VEGFR-2 leads to degradation of the basement membrane followed by the transformation of ECs to motile tip cells at the sprouting end of the vessel [87,88] endothelial cell migration [89], and EC proliferation [90,91,92]. Angiogenesis is mediated by the coordination of dynamic tip cells with filopodia and stalk cells. Tip cells sense the pro or antiangiogenic factors in the tumor microenvironment and migrate, while the stalk cells have less filopodia and a faster proliferation rate to facilitate tube formation [88]. The new ECs establish tight junctions, and hence, a luminal vessel is produced [93,94]. The newly formed vessels are then covered by tumor-recruited pericytes, which provide support to the vessels from outside (maturation step) (Figure 3) [95]. Unlike normal blood vessels, blood vessels in tumors are more permeable and tortuous, which is attributed to the altered endothelial cells (ECs) referred to as Tumor Endothelial Cells (TECs) [96,97]. Furthermore, besides the autocrine production of proangiogenic factors by tumor cells, tumor cells can prime immune cells to produce more angiogenic factors, which results in a loss of balance between the pro and antiangiogenic factors, rendering the tumor vasculature unruly and poorly developed [98]. Tumour blood vessels thereby have loose junctions, an incomplete basement membrane distribution, and reduced pericyte support. This deformed vascular physiology leads to poor blood supply and, therefore, hypoxia and reduced drug delivery [24,99].

Tumor vasculature and angiogenesis are important prerequisites of tumor cell proliferation, survival, and progression. Therefore, it is a feasible approach to inhibit tumor angiogenesis to target cancer proliferation. In the early 1990s, a murine anti-human VEGF-A antibody was developed that could reduce angiogenesis in vivo. A few years later, the antibody was humanized and was termed Bevacizumab (Avastin), which had the ability to bind and neutralize all VEGF-A isoforms [101]. Bevacizumab was approved by the U.S. Food and Drug Administration (FDA) as a first-line antiangiogenic therapy for colorectal cancer (CRC) in 2004 and is a common drug of choice for various cancers today [102]. Ever since the discovery of Avastin, tumor angiogenesis and factors that might be involved in the regulation of the process have been extensively researched. There have been several candidates, but only a few were approved for antiangiogenic therapy to treat cancer, age-related macular degeneration (AMD), and diabetic retinopathy. Approved VEGF-A specific monoclonal antibodies for the treatment of cancer include Bevacizumab (Avastin for non-small cell lung cancer (NSCLC), glioblastoma, metastatic renal cell-, and cervical cancer) and ramucirumab (Cyramza for gastric, NSCLC and metastatic colorectal cancer). Further anti-VEGF-A antibodies are approved for macular edema (Ranibizumab (Lucentis) and Aflibercept (Eylea)). In addition, the targeted therapy, antibodies with a broader target range targeting tyrosine kinase receptor, i.e., sorafenib (Nexavar), regorafenib (Stivarga), and sunitinib (Sutent) are also used to inhibit VEGFR activation and hence angiogenesis [103,104].

### 3.1. VEGF-D and Blood Capillary Angiogenesis

Like many other types of anticancer treatments, tumor cells developed resistance to antiangiogenic therapies [105,106,107]. Altered gene expression of angiogenesis-related genes in tumor cells was demonstrated in response to antiangiogenic drugs. Particularly, the expression of VEGFs was altered both in tumor cells and the surrounding stroma [103,107,108,109]. Since other members of the VEGF family could activate VEGFR-2 in the absence of VEGF-A, this could be a potential mechanism of antiangiogenic therapy resistance [110]. An interesting finding was that Avastin-mediated VEGF-A inhibition led to an upregulation of VEGF-D expression in gliosarcomas, and although the tumor growth was slower, the tumors when established, had better vascularization [111]. The normalization of this neo vasculature post-VEGF-A inhibition resulted in morphological changes: increased pericyte coverage, less leaking, less dilation and functional changes; decreased interstitial fluid pressure, increased tumor oxygenation, and improved penetration of drugs into these tumors [112]. VEGF-D can, therefore, execute all steps of tumor angiogenesis in the absence of VEGF-A with comparable efficiency [91,100] (Figure 3). 

Similarly, upregulated VEGF-D expression was reported in inflammatory breast cancer after treatment with celecoxib (non-steroidal anti-inflammatory drug) and VEGFR-2 inhibitor SU5416 [41]. Furthermore, patients with nonresectable hepatocellular carcinoma, already receiving bevacizumab, had a higher VEGF-D expression and promoted disease progression [113]. The ability of VEGF-D as an inducer of blood capillary angiogenesis was first characterized in rabbit retinal tissue [114] and was confirmed in human embryonic kidney cell line: induced VEGF-D expression resulted in the formation of highly vascularized and non-oedemic tumors as compared to tumors of wild type cells in mice [115]. VEGF-D can induce CD-31, as well as LYVE-1 positive vessels, by activating either VEGFR-2 or VEGFR-3, respectively, in the absence of VEGF-A [115]. Upon adenoviral vector delivery to skeletal muscles, mature VEGF-D was an excellent inducer of both lymphangiogenesis and angiogenesis, promoting pericyte recruitment and vascular permeability and a more diffused angiogenesis pattern [116]. A similar dose-dependent angiogenic response was observed in pig hearts [117]. VEGF-D also induced angiogenesis in the brain after blood–brain barrier breakdown by activating both VEGFR-2 and VEGFR-3 [118]. VEGF-D modulates angiogenesis by inducing EC proliferation and migration to branching points and facilitating tube formation (Figure 3) [116,119,120]. Based on these findings, VEGF-D expressed by tumor cells could be a way for cancer to circumvent VEGF-A targeting antiangiogenic therapy [121,122,123]. Interestingly, other forms of anticancer therapies, such as docetaxel and vinorelbine, also upregulate VEGF-D expression and induced angiogenesis in vitro in HUVEC and BC cells in relation to melatonin signaling [124]; this further indicates that VEGF-D has a unique yet unidentified expression and signaling pattern which should be explored to facilitate multi-targeted antiangiogenic therapy. 

### 3.2. VEGF-D Mediated Angiogenic Signaling in Cancer

Since VEGF-D can activate VEGFR-2 mediated signaling, it mainly activates signaling pathways involved in the proliferation, migration, survival, and physiology of EC cells [104], essentially maintaining the vascular system in development and disease [23]. Although researchers are in agreement that upregulated VEGF-D in response to antiangiogenic drugs might be one of the key factors enabling antiangiogenic drug resistance in cancers [103], little is known about the signaling pathways and expression regulators involved. Here, we are mentioning data available to date about VEGF-D signaling in angiogenesis.

One of the main factors promoting VEGF-D expression in response to antiangiogenic therapy would be therapy-related hypoxia [125]. Achen et al. reported that VEGF-D was secreted exclusively by tumor cells in two independent model systems, acting in a paracrine fashion to activate VEGFR-2 positive blood vessels in tumors [126]. Furthermore, VEGF-D secreted by the tumor cells was localized in metastatic melanoma blood vessels endothelial cells near the immune-positive tumor cells but not in the distant blood vessels, which further supports that VEGF-D promotes angiogenesis in a paracrine fashion [55,127,128]. However, due to the localized production of VEGF-D by tumors and its role in promoting tumor cell progression, migration, and metastasis, it might also have an autocrine function in addition to its paracrine signaling as reported in endometrial carcinoma [129] and invasive cervical carcinoma [130]. Furthermore, VEGF-D could induce proangiogenic phenotype in HUVECs by upregulating genes involved in matrix modulation and cell membrane alteration in an autocrine manner [131]. Interestingly, in Gastric cancer (GC), VEGF-D transcription was facilitated by proteolytic activation of transcriptional factor CDP/Cux p200 by protease cathepsin L (CTSL). This upregulated VEGF-D translated to higher angiogenesis in GC [132]. Recently, in GC, a long noncoding RNA (LncRNA) CRART16 was found to downregulate miR-122-5p, thereby upregulating transcriptional factor FOS and hence upregulated VEGF-D expression and angiogenesis both in vitro and in vivo [133]. Based on these findings, we believe the autocrine action of VEGF-D might be stronger in executing angiogenic response and a potent target to manage antiangiogenic drug resistance.

Some interesting points for research would include investigating if VEGF-D activates the same downstream signaling cascades as VEGF-A when activating VEGFR-2. Based on the bioinformatics analysis, VEGF-D potentially activates the MAPK signaling pathway, promoting cell proliferation, differentiation, and survival [134]. Secondly, as in glioblastoma, vascular, breast, and lung cancer, the tumor tissue has high expression of VEGFR-3 that unconventionally contributes to maintaining endothelial integrity in tumor blood vascular angiogenesis [135,136,137,138]. It would be interesting to investigate how VEGF-D/VEGFR-3 signaling is stimulating angiogenesis. 

## 4. Clinical Significance of VEGF-D in Tumor Angiogenesis

Based on Cancer Protein Atlas data, VEGF-D expression is prevalent in malignant melanomas, urothelial, gastric, and pancreatic cancers [139]. VEGF-D expression is clinically detected by its serum and plasma levels by Enzyme-linked immune-sorbent assay (ELISA), multiplex assays such as Luminex technology [140], or by immunohistochemistry in resected tumor tissue. Although the mechanistic basis of VEGF-D signaling in cancer angiogenesis has not been fully elucidated, patient data analysis has indicated its significance in cancer progression and prognosis. In lung cancer, IL-6 promoted tube formation by enhancing VEGF-D expression [141]. Similarly, in invasive ductal breast carcinoma, VEGF-D but not VEGF-A or VEGF-C was upregulated in tumor tissue and promoted angiogenesis [142]. Cell lines established from Tumor-derived Endothelial Cells (TEC) from renal carcinoma patients expressed VEGF-D and VEGFR-2 but not VEGF-A compared to normal human mammary epithelial cells in vitro. Moreover, TEC implants in matrigel in SCID mice confirmed VEGF-D expressing TECs in vivo [143]. In pediatric hepatoblastoma and hepatocellular carcinoma, VEGF-D secreted by the reactive ductules (cells with a ductular phenotype that proliferate and accumulate in response to liver damage) was associated with a higher number of CD34+ (a marker of proliferating endothelial cells), blood vessels compared to low VEGF-D expressing hepatic tumors [144]. Therefore, the unique receptor expression of tumor vasculature and VEGF-D’s ability to activate more than one of them makes it a prominent regulator of blood vessel angiogenesis. Furthermore, these results are indicative of VEGF-D’s ability to circumvent antiangiogenic therapy and promote cancer survival and progression.

Clinical and pathological data established the role of VEGF-D in cancer, as high VEGF-D expression is linked to poor outcomes. Increased VEGF-D expression is an independent negative prognostic marker in ovarian carcinoma, colorectal cancer, breast cancer, and gastric carcinoma [145,146,147,148,149]. Furthermore, in different grades of endometrial cancer, a higher VEGF-D expression was observed in correlation to a higher tumor grade [150]. Similarly, a pilot study reported an increase in VEGF-D in early epithelial ovarian tumors, potentiating its role as an early-detection biomarker [151]. Clinical studies for metastatic colorectal cancer (mCRC) revealed that targeting VEGF-A with bevacizumab upregulated VEGF-D expression, explaining a poor response to bevacizumab [152,153]. However, mCRC patients with high VEGF-D expression, when treated with VEGFR-2 inhibitor ramucirumab, had a higher overall (OS) and progression-free survival (PFS) [154,155]. Higher VEGF-D expression also translated to poor prognostic significance and a 3-fold higher death risk in CRC patients [156,157,158]. In another clinical trial investigating mCRC in bevacizumab-treated patients, high plasma VEGF-D was associated with poor clinical outcomes [159]. Very recently, a phase II clinical study also showed similar results when a VEGFR inhibitor, Dovitinib, increased plasma VEGF-D levels [160].

Bioinformatics analysis of patient data from cancer databases of hepatocellular carcinoma (HCC) samples identified VEGF-D as a negative prognostic angiogenic marker [134,161]. VEGF-D was also upregulated in response to radiotherapy in lung cancer and was interpreted as an angiogenic biomarker [162]. Taniguchi and his colleagues’ clinical trial also reported that VEGF-D expression could be induced by Bevacizumab treatment; hence, its baseline expression can help choose a better antiangiogenic treatment regimen [163]. Similarly, a clinical trial investigated the efficacy of VEGF-D as a predictive biomarker prior to initiation and during second-line treatment with paclitaxel and ramucirumab in GC patients [164]. Furthermore, another clinical trial found a higher PFS in mCRC patients with lower VEGF-D compared to patients with high VEGF-D plasma levels when treated with ramucirumab and no previous bevacizumab therapy [140]. There is substantial evidence that VEGF-D has significant clinical relevance in antiangiogenic therapy. However, the mechanisms upregulating VEGF-D expression and downstream signaling activated by VEGF-D are still unclear. We believe a thorough insight into how VEGF-D steps up as an angiogenic mediator is an interesting research question with substantial translational potential.

## 5. Current VEGF-D Therapies

Despite its importance, there is currently no specific inhibitor of VEGF-D. Reasons include:(1)The structural similarity between VEGF-D, VEGF-C, and VEGF-A with similar receptor-binding domain (VHD);(2)The complex VEGF-D signaling through VEGFR-2, VEGFR-3, and NRP-1;(3)The multiplex proteolytic cleavage of VEGF-D by different proteases providing different isoforms with variable structures.

Furthermore, since the role of VEGF-D has been established in lymphangiogenesis and is deemed secondary in tumor blood neoangiogenesis, less attention has been devoted to the development of specific VEGF-D inhibitors.

VEGF-D can be directly targeted by monoclonal antibodies (mAb). Achen et al. generated antibodies against the VHD of various VEGF-D isoforms. VD-1 effectively reduced the interaction of human VEGF-D with both VEGFR-2 and VEGFR-3 and inhibited microvascular epithelial cells’ mitotic response to VEGF-D [165]. Further studies by Davydova et al. included protein mapping of the epitope binding site of VD-1 and reported that VD-1 binds to and inhibits a specific amino acid region ‘147NEESL151’ of VEGF-D to neutralize its action. However, a mutation in this particular amino acid sequence renders VD-1’s activity ineffective [166]. They also compared the effectivity of VD-1 with commercially available mAb-286, which binds the alpha helix structure of mature VEGF-D, and demonstrated that, like VD-1, mAb-286 is also ineffective in inhibiting receptor binding action of all VEGF-D variants [167]. It is, therefore, difficult to design an antibody that would neutralize all pre and post-translational variants of VEGF-D. Thus, further current treatment modalities include targeting VEGF-D inducers, such as c-fos, or inhibition of VEGF-D’s receptor interaction and/or receptor activation.

Ji and colleagues applied andrographolide, a plant-based naturally occurring diterpenoid lactone, which reduced c-fos expression and nuclear translocation, thereby resulting in reduced VEGF-D expression and angiogenesis in hepatoma cancer cells [168]. Inactivation of VEGF-D function by hindering receptor interaction and activation is another widely used approach that has yielded positive results. VGX-300: a soluble VEGFR-3 successfully reduced VEGF-D activity in age-related macular degeneration and diabetic macula edema of the eye in a rat model [169,170]. Similarly, a monoclonal antibody against VEGFR-3, mF4-3C1, could reduce lymphatic regeneration in mice [171], while a humanized antibody hF4-3C5 could reduce tube formation in response to VEGF-D by 60% in bovine aortic endothelial cells [172]. More recently, in a clinical trial, promising results were observed in age-related macular degeneration neovascularization by using VEGF-C/VEGF-D inhibitor OPT-302 along with VEGF-A inhibitor ranibizumab. OPT-302 is a trapping molecule composed of VEGFR-3 ligand binding domains that bind and reduce the availability of VEGF-C and VEGF-D for the endogenous VEGFR-3 receptor [173]. An ongoing clinical trial is aiming at evaluating the safety and pharmacological properties of OPT-302 in advanced neovascular age-related macular degeneration [174]. Similarly, monoclonal antibodies such as sorafenib, ramucirumab, etc., block the VEGFR-2 ligand interaction and can be used to target VEGF-A or VEGF-D mediated signaling. These VEGFR-2 mAbs reduce VEGF-D’s ability to activate VEGFR-2 and, thus, can inhibit the angiogenic role of VEGF-D. Tanibirumab, a humanized antibody against the VEGF binding domain of VEGFR-2, was very successful in blocking the receptor activation by VEGF-C and VEGF-D with fewer adverse effects [175]. Currently, two clinical trials are investigating the safety and tolerability of Tanibirumab in advanced metastatic cancer and recurrent glioblastoma [176,177]. These approaches, although promising, are nonspecific for VEGF-D and thus are not suitable to dissect the relative roles of VEGF-A or VEGF-D. The antibiotic salinomycin effectively reduced VEGF-D expression in endometrial cancer cells, as reported recently in 2021 [178]. Similarly, a naturally occurring small compound used in Chinese medicine, norcantharidin, also effectively reduced VEGF-D mRNA production [179].

The tumor vasculature development has a complex regulation and, thus, offers several points to interfere with tumor development, but requires a carefully crafted treatment regimen to halt the potential crosstalk between VEGF family members to have maximal effects. The most used treatment modalities at present to reduce VEGF-D activity are summarized in Table 1.

## 6. Conclusions

VEGF-A is well-established as a regulator of cancer angiogenesis, and substantial clinical evidence has accumulated already with approved inhibitors supporting anti-VEGF-A therapy in cancer treatment. There is accumulating evidence that VEGF-D, besides its role in regulating lymphangiogenesis, is also an important angiogenic factor, and can mediate angiogenesis in a manner comparable to VEGF-A. Although initially, VEGF-D stepping up as an angiogenic factor was detected in response to VEGF-A inhibition, later findings established VEGF-D as an angiogenesis biomarker in cancers independent of VEGF-A inhibition. Furthermore, VEGF-D may have direct effects on cancer cell proliferation, rendering it a primary anticancer therapeutic target. Substantial clinical evidence supports the therapeutic potential of VEGF-D inhibition; however, the lack of specific inhibitors at present prevents the dissection of the sole role of VEGF-D from the complex inhibition of the VEGF family by these inhibitors.

## Figures and Tables

**Figure 1 ijms-24-13317-f001:**
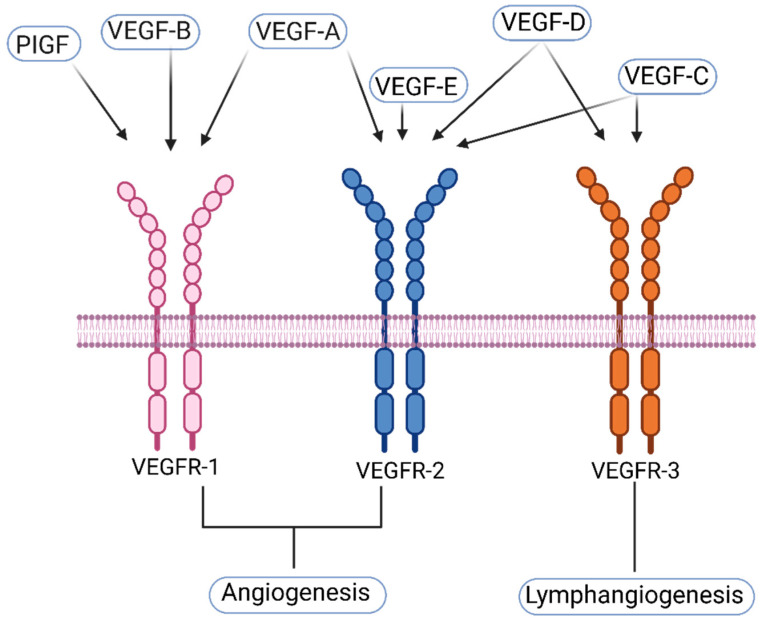
VEGF-receptors and their ligands. VEGF-A binds VEGFR-1 and 2. VEGF-B and PIGF both bind VEGFR-1 only. VEGF-C and -D can bind both VEGFR-2 and 3. VEGF-E binds only VEGFR-2. VEGFR-R1 and 2 regulate blood vessel angiogenesis, whereas VEGFR-3’s main function is the regulation of lymphangiogenesis. VEGFR-1 also acts as a decoy receptor, preventing its ligands from binding VEGFR-2 [7,8,10,11,12]. Created with BioRender.com accessed on 27 July 2023.

**Figure 3 ijms-24-13317-f003:**
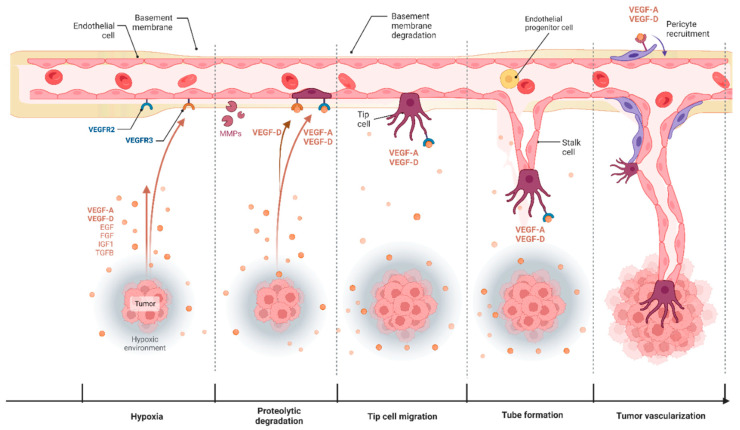
Steps of tumor angiogenesis. Created with BioRender.com accessed on 27 July 2023 based on: [91,100]. Step 1 (hypoxia): the hypoxic or inflamed tumor microenvironment (TME) induces the production of vasculogenetic growth factors. Step 2 (proteolytic degradation): VEGF-A and -D binding to VEGFR-2 induce the production of matrix metalloproteinases (MMPs), degrading the extracellular matrix (ECM). Step 3 (tip cell migration): VEGFR-2 activation induces the transformation of ECs to tip cells (brown). Step 4 (tube formation): tip cells migrate into the TME with coordinated proliferation of stalk cells, forming a new tube. Step 5 (tumor vascularization): the new vessels mature by establishing tight junctions and by the coverage by pericytes (lila).

**Table 1 ijms-24-13317-t001:** Antiangiogenic drugs targeting VEGF-D.

Drug	Mechanism of Action	Selectivity, Mechanism of Action	Reference
Andrographolide	Targets the VEGF-D inducer: c-fos	Non-selective	[168]
VGX-300	Soluble VEGFR-3	Bind and inhibit the activation of VEGFR-3 by all of its ligands (VEGF-D/C/A)	[169,170,171,172,173,180,181,182]
SAR131675	VEGFR-3 tyrosine kinase inhibitor (it does not say about its property. Inhibits phosphorylation and hence activation of the receptor)		
mF4-31C1, hF4-3C5	mABs against VEGFR-3		
OPT-302	VEGFR-3 decoy		
mAB 286	mAb	Selective: Binds mature VEGF-D, prevents its binding to VEGFR-2 and VEGFR-3	[165,167]
VD1	mAb	Competes with VEGF-D in activation of VEGFR-2 and VEGFR-3	[166]
Tanibitumab, Ramucirumab, SU5614, Sunitinib, Benzoxazole and its derivatives	VEGFR-2 inhibitors	VEGFR-2 inhibition prevents blood angiogenic signaling from (VEGF-A/C/D)	[57,183,184,185,186]
Endostatin	VEGF-D (and VEGF-A, VEGF-C) inhibitor	Non-selective	[187]

## Data Availability

Data sharing is not applicable to this article.

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
