# Peer review of "Vascular Endothelial Growth Factor-D (VEGF-D): An Angiogenesis Bypass in Malignant Tumors"

_ijms, 2023, doi:10.3390/ijms241713317_

Round 1

Reviewer 1 Report

The authors in this manuscript entitled " VEGF-D: an angiogenesis bypass in malignant tumors" has demonstrtaed generally about VEGF mechanistics and their types and targets. I have the following comments.

1. I suggest for the authors to either change the title or adding more about research discussinf role of VEGF-D in cancers. 

2. Only sections 4 and 5 discussing the targets and clinical aspects. I suggest to add mor clinical trials table or adding current VEGF-D drug target prospective research.

3- I suggets to proofread the language of the manuscript. There are many errors existed. 

4- Adding future direction of research on VEGF-D will add benefits to this review. 

5- There are previous report about other innovation in angiogenesis terapy. you can take them as example: https://doi.org/10.3390/ijms22041631 

Minor changes can be done for English writing for this manuscript. 

Author Response

1. I suggest for the authors to either change the title or add more about research discussing the role of VEGF-D in cancers.

We thank the reviewer for their suggestion, however, we believe the title provides a concise insight into the message we intend to put forward about the role of VEGF-D as an angiogenic factor. Unfortunately, there is not enough data available on VEGF-D as an angiogenic factor. However, we believe the available evidence is quite strong to investigate VEGF-D’s role in tumor angiogenesis, therefore we have summarized all available data in this paper.

2.Only sections 4 and 5 discuss the targets and clinical aspects. I suggest adding more clinical trial tables or adding current VEGF-D drug target prospective research.

There is only a single clinical trial undergoing which is investigating the role of VEGF-D as a predictive biomarker in cancer in relation to angiogenesis. We also tried to update the current therapies section however we did not come across any new targeted therapy against VEGF-D. We have already mentioned the unavailability of targeted anti-VEGF-D therapy in our manuscript.

3. I suggest to proofread the language of the manuscript. There are many errors existed.

We have proof-read the manuscript again and have corrected errors.

4. Adding future direction of research on VEGF-D will add benefits to this review.

In the revised manuscript we have added future directions and potential research questions in several sections.

5. There are previous report about other innovation in angiogenesis terapy. you can take them as example: https://doi.org/10.3390/ijms22041631

We thank the reviewer for their kind suggestion.  We took inspiration and restructured the paper by making subheadings and rearranging the text enabling clarity for the readers.

Reviewer 2 Report

The overall evaluation for the manuscript “VEGF-D: an angiogenesis bypass in malignant tumors” is just satisfying; 

The paper  is not very innovative.

Author Response

1. The paper is not very innovative.

We thank the reviewer for their comment. The manuscript provides a unique insight into VEGF-D being a potentially stronger angiogenic factor than VEGF-A. The manuscript is the first paper to date that has summarized all available data on VEGF-D being able to execute angiogenesis both in-vitro and in-vivo, its clinical relevance and potential therapies against VEGF-D. We believe the paper is highly innovative as it provides a fresh perspective on the role of VEGF-D as an angiogenic factor. We hope the reviewer reads the paper with an open mind and acknowledges the way we have organized available literature and findings to help plan future research on VEGF-D as an angiogenic bypass.

Reviewer 3 Report

The review article entitled “VEGF-D: an angiogenesis bypass in malignant tumors” it is an interesting paper bringing together an interesting content regarding VEGF-D in clinical practice. The figures are interesting and well-designed. Please, see specific comments below:

1.     VEGF could be increased due autocrine and paracrine tumor interactions. Authors described a little bite about this topic, but could explore more. According to the authors article, seems that autocrine production should be more specific to different treatments?

2.     Authors also could explore more the recent clinical trials focused on VEGF-D and angiogenesis. There is any current clinical trial being made?

3.     Authors also could explore we can explore the knowledge of VEGF-D expression in clinical routine. Should we perform IHC or PCR to evaluate its expression? How select patients for the therapy?

4.     Author also could summarize which tumors subtype is more linked with VEGF-D overexpression related to angiogenesis.  

Author Response

1. VEGF could be increased due to autocrine and paracrine tumor interactions. The authors described a little bit about this topic but could explore more. According to the author’s article, seems that autocrine production should be more specific to different treatments.

We thank the reviewer for their kind suggestion and updated the manuscript by adding more information about autocrine and paracrine signaling in section 3.2. We believe targeting autocrine signaling would have more robust effects.

2. Authors also could explore more the recent clinical trials focused on VEGF-D and angiogenesis. There is any current clinical trial being made?

We thoroughly checked the clinical trial data (at clinicaltrials.gov) investigating VEGF-D in relation to angiogenesis and found only one clinical trial investigating VEGF-D as a predictive biomarker in relation to angiogenesis which we have described in the updated manuscript.

3. Authors also could explore we can explore the knowledge of VEGF-D expression in clinical routine. Should we perform IHC or PCR to evaluate its expression? How select patients for the therapy?

We thank the reviewer for pointing out these questions. We have updated the manuscript and mentioned the current methods employed for the detection of VEGF-D and that early serum and plasma level detection can give insight to plan treatment regimens.

4. Author also could summarize which tumors subtype is more linked with VEGF-D overexpression related to angiogenesis.

We have added data from the Human protein Atlas and other publications to mention the prevalence of VEGF-D.

Round 2

Reviewer 1 Report

Dear authors,

You have address all the suggested comments. 

One minor comment about heading sections.

1. Introduction should be separate heading

2. will be other parts and start subheading 

I suggest to accept the manuscript 

None. 

Reviewer 2 Report

Several anti-angiogenic drugs (including anti-VEGF-A and/or drugs to block its receptors) have been approved for cancer treatment, alone or in combination with other anti-tumoral agents, howewer, very few evidences highlights the importortance to block tumor angiogenesis via VEGF-D, thus this manuscript seems, partially, cover these gaps.

The title of the manuscript seems now intriguing for both clinical and research readership.

The paper is supported by experimental studies an by appropriate bibiography,

The figures and the tables are adequately representative and well supported by main text and by their appropriate captions.

To conclude,  the overall evaluation of the revised version of the manuscript "VEGF-D: an angiogenesis bypass in malignant tumors" is satisfying.